# Classification of the Residues after High and Low Order Explosions Using Machine Learning Techniques on Fourier Transform Infrared (FTIR) Spectra

**DOI:** 10.3390/molecules28052233

**Published:** 2023-02-27

**Authors:** Agnieszka M. Banas, Krzysztof Banas, Mark B. H. Breese

**Affiliations:** 1Singapore Synchrotron Light Source, National University of Singapore, 5 Research Link, Singapore 117603, Singapore; 2Department of Physics, National University of Singapore, 2 Science Drive 3, Singapore 117542, Singapore

**Keywords:** high and low order explosions, machine learning techniques, Fourier Transform Infrared (FTIR) spectroscopy, spectral analysis, explosive residues

## Abstract

Forensic science is a field that requires precise and reliable methods for the detection and analysis of evidence. One such method is Fourier Transform Infrared (FTIR) spectroscopy, which provides high sensitivity and selectivity in the detection of samples. In this study, the use of FTIR spectroscopy and statistical multivariate analysis to identify high explosive (HE) materials (C-4, TNT, and PETN) in the residues after high- and low-order explosions is demonstrated. Additionally, a detailed description of the data pre-treatment process and the use of various machine learning classification techniques to achieve successful identification is also provided. The best results were obtained with the hybrid LDA-PCA technique, which was implemented using the R environment, a code-driven open-source platform that promotes reproducibility and transparency.

## 1. Introduction

The forensic identification of explosives is a topic of considerable interest to all branches of homeland security due to the constant threat of increasing criminal activities. Trace residues of explosives have a high evidentiary value as they represent the chemical composition of the material used in the explosion and could provide information on whether it was a homemade or commercial product. In some cases, this information can also lead to the identification of the criminal organization [1]. There are many papers that describe the analysis of high-order explosive materials using different approaches, but the analysis of materials left after explosions is still a topic that is not often discussed by scientists, even though it could potentially help in the identification of the explosives that caused the blast [2]. The reason for this is clear: the post-explosion scenario usually portrays a place of chaotic destruction, with limited amounts of substances that could potentially be treated as evidence of high explosive use. Moreover, the optimal location for collecting traces of unreacted explosives has not been scientifically determined. There are some guidelines for crime scene investigation [3], but not many scientific papers. The amount of evidence increases when the explosion occurs in confined spaces, such as mines, buildings, or large vehicles. However, it is not always the case, as an enclosed area in which the bomb is placed can contribute to an increase in the detonation pressure. Parameters such as high pressure and high temperature can speed up the mixing process of reaction products and lead to their wider dispersion in the air [4]. The efficiency and type of the explosion are also important parameters used by trained personnel to describe the explosive event. A low-order explosion occurs when the blast pressure front moves at a slow pace and displaces or distorts objects in its path. From the perspective of terrorist attacks, low-order detonations usually occur as unintended incidents, or malfunctions of ordnance that result in a significantly lower yield than designed [5]. Very often, the main reason for an incomplete explosion may be the usage of old or deteriorated explosives, a failure in the booster or detonator [6] or some inhomogeneity in the primary charge [7,8]. In a low-order explosion, a quite high quantity of non-reacted particles from explosives is left on the surface of the remnants, as the consumption of the explosive is less than 100%. With high-order explosion, only scarce amounts of post-blast residues are detected, as detonation is characterized by the rapid combustion of explosive materials, a rapid expansion of the resultant gases, and a build-up of high heat and pressure. The blast pressure front moves rapidly, shattering the objects in its path [9]. According to the literature, undetonated residues are found not only after low- but also after high-order explosions [10,11]. Factors such as the velocity of detonation, charge mass, charge diameter, and the number of interfaces are responsible for that fact. Correct forensic identification of the post-blast residues depends on the use of proper analytical methods, as well as the analyst’s expertise. Fourier Transform Infra-Red (FTIR) Spectroscopy [12,13,14], Raman Spectroscopy [15,16], X-ray Diffraction (XRD), Gas Chromatography combined with Mass Spectrometry (GC-MS) [17], Gas Chromatography combined with Thermal Energy Analyzer (GC-TEA or EGIS) Spot Test, Liquid Chromatography-Mass Spectrometry (LC-MS) and Energy Dispersive X-ray Analyzer (EDX) are adopted as the primary techniques producing meaningful chemical information from tested samples [18,19]. FTIR spectroscopy is one of these techniques that require for a tiny amount of sample for an analysis and allow for its fast identification. Upon shining infrared light, a portion of the incident radiation at a specific energy is absorbed by the specimen. Because various chemical functional groups absorb infrared light at exact wavelengths, the resulting FTIR spectrum can be considered the molecular fingerprint of the sample. FTIR spectroscopy is a powerful analytical technique that is widely used in many laboratories worldwide to analyze various types of samples. In this work, post-blast residues refer to the unreacted microscopic particles of explosives that remain intact after the explosion. The analysis of post-blast residues by FTIR spectroscopy was the subject of our previous papers [20,21]. In this work, we focus on implementing machine learning classification techniques that should allow us to successfully identify explosive materials even in the remnants after high-order explosions. It is clear that visual inspection of collected FTIR spectra is not adequate for the assessment process and does not support the recognition of the relationship between various chemicals and their IR fingerprints. Therefore, alternative approaches are needed. To achieve this systematically and reproducibly, we need a tool that allows for the processing and analysis of results with concurrent control of applied procedures and selected parameters. The R environment [22] for statistical analysis is an extremely versatile platform for scientific data evaluation. The best graphic user interface (GUI) for this environment at present is RStudio [23]. The power and flexibility of R lies in its modular structure, which allows the mounting of required packages (libraries) and code-driven data analysis approaches at any time. The research gap in the area of forensic identification of explosives lies in the lack of scientific investigation on the analysis of post-explosion residues. Further research is needed to systematically and reproducibly analyze these residues in order to aid in the identification of explosive materials.

The main objective in our work is to develop a reliable and efficient method for identifying and classifying the residues produced by high- and low-order explosions. This will be achieved through the use of FTIR spectroscopy combined with machine learning techniques to analyze the spectral data collected from the residues. We would like to emphasize that, to the best of our knowledge based on the available literature, the results presented in this paper are the first of their kind in this field. This makes our work particularly valuable and relevant to the research community dealing with the analysis of post-blast residues.

## 2. Materials and Methods

Post-blast remnants were obtained after controlled explosions performed by the specially trained staff at the Advanced Materials Engineering Pte Ltd (AME) facility in Singapore. Three high-explosive materials (C-4, PETN, and TNT) were selected to undergo the exercises. Additionally, two types of explosions were carried out: low- and high-order to evaluate the impact of the explosion strength on the number of remnants and to assess our ability to identify the high-explosive materials. Various everyday objects, including glass, steel, plastic bag, plywood, chipboard, cardboard, hose pipe, towel, and fabric, were used during the explosions. These objects were attached to a perforated steel cage designed to fit inside a small steel container (18 by 18 by 18 cm${}^{3}$). Some materials were also placed at the bottom of the container to increase the number of samples for analysis. After the explosion, the proximity of the container was initially searched for, as it is known that explosive residues decrease in concentration with increasing distance. All potentially useful objects for analysis were labeled and stored in separate foil bags. Most of the small sample catchers were found inside the container even after a high-order explosion. Generally, the place after the high-order explosion appeared messier compared to the situation after the low-order explosion. With the low-order blast, macroscopic particles of the used explosive materials were still visible on the surfaces of objects. During the sample preparation process, they were gently removed and mixed with potassium bromide (KBr) in a weight ratio of around 1:100. The homogenous powder was pelletized with the usage of a hydraulic press with a clamping force of 80 kN. Thin, 13 mm in diameter pellets were placed in the sample holder, inserted into the sample chamber of FTIR spectrometer IFS66v/S, and analyzed in the transmission mode. Before the experiments, the sample chamber was evacuated down to 3 mbar to eliminate the content of the water absorption lines in the final spectrum. The spectra were collected within the 4000–400 cm${}^{-1}$ range using a nominal spectral resolution Δν= 4 cm${}^{-1}$ (data spacing of 2.04 cm${}^{-1}$), and 67 scans (1 min of experiment time) were averaged to obtain good signal-to-noise ratio spectra. Each pellet was analyzed at several points to determine the reproducibility of the results. During all measurements, a Mercury Cadmium Telluride (MCT) detector cooled to liquid nitrogen temperature (77 K) was used. A background was collected in an empty (also evacuated to 3 mbar) sample chamber as a reference to the single beam intensity and the effect of atmospheric changes. For samples that survived high-order blasts and showed visual signs of damage, such as cratering or pitting, a cotton swab soaked in acetone was used to clean the surface or the samples were thoroughly rinsed with acetone. The same approach was taken with samples that showed no detectable signs of an explosion, as in such cases unseen traces of explosive residues can still be present on their surfaces. The solvent was transferred to an agate mortar, and after acetone evaporation, the remaining products were mixed with KBr. The KBr pellet was analyzed using the procedure described earlier. Over 200 samples containing residue materials gathered from various surfaces of objects found in the steel container or in close proximity to the center of the explosion were prepared. Intact high-explosive materials (C-4, PETN, and TNT) have also been analyzed and included in our reference spectral database. Spectral data were collected using the native Bruker Opus software and exported to ASCII files through a macro procedure. All files were then imported into R software version 4.2.2 [22]. R Studio version 2022.07.2 [23] was used as the graphical user interface for the R language. In this data evaluation protocol, the following libraries (packages written for the R Environment) were used: hyperSpec [24] for hyperspectral datasets manipulations, ggplot2 [25] for high-quality visualizations, caret [26] for the evaluation of the classification (confusion matrices), MASS [27] for function LDA, klaR [28] for 2D partition plots, FactoMineR [29] and factoextra [30] for PCA-related calculations and visualizations, and finally tidyverse [31] for the data manipulation.

## 3. Results

Prior to any further analysis, careful data pre-treatment was applied to reduce and eliminate possible sources of errors, to reduce noise and enhance essential signals, to correct the sloped or oscillatory baselines due to scattering effects, and to remove the signals derived from atmospheric water vapor, carbon dioxide or other interfering compounds [32]. The baseline was calculated as the least squares polynomial of order 6; the appropriate support points for the baseline were found iteratively. It is known that pre-processing steps are essential for data analysis; however, there is no clear rule on which algorithm should be followed. Special attention must be paid to selecting the proper scheme. Several normalization methods (including min–max, vector normalization, 1-norm and area normalization) have been tested to lessen the spectral variability and overcome the confounding effect of varying sample thicknesses on the band intensity [32]. In our work, area normalization (division by the mean value for each spectrum) was selected based on the value of the $\chi^{2}$ metric [33]. Preliminary analysis based on a comparison of the collected FTIR spectra for intact high-explosive materials (C-4, PETN and TNT) with spectra found in the commercially available database confirmed that all lines characteristic for the selected HEs were detected by our spectrometer. The FTIR spectra collected for post-blast residues are very multifaceted, as they contain not only traces of explosive materials, but sometimes also information about the chemical composition of samples that were used as post-blast residue catchers. To obtain meaningful information allowing for proper classification of high explosive materials found in those spectra, multivariate statistical methods are needed. As was mentioned earlier, all transmission spectra were collected within the spectral range of 4000 to 400 cm${}^{-1}$, which means that the number of variables for the set of available spectra was very high (1765 wavenumbers = variables). That is why we have decided to initially reduce this number by selecting only the fingerprint region for further analysis. Before that, the influence of IR regions on HEs identification was examined. PCA (principal component analysis) as an unsupervised pattern recognition method was applied to FTIR spectra collected for intact HE materials in three regions (1900–400 cm${}^{-1}$, 4000–400 cm${}^{-1}$ and 3000–2800 cm${}^{-1}$). It was carried out to visualize in a simple way how similar or different the spectra within these regions are. Prior to PCA, all regions were separately subjected to normalization for better enhancing even lower intensity bands. All PCAs in this work were completed employing a non-linear iterative partial least squares (NIPALS) algorithm, cross-validation with an uncertainty test, and 1/SDev as weighing. As was expected, the fingerprint region (1900–400 cm${}^{-1}$) containing the most characteristic absorption lines was sufficient in unequivocal spectra separation. It is due to the fact that the same molecules contribute to stretching and bending modes, so not only the “silent region” (2700–1900 cm${}^{-1}$), but also a higher wavenumber region, i.e., 4000–2700 cm${}^{-1}$, having only a few characteristic lines, can be eliminated from further analysis [34]. To conclude, all further analyses presented in this paper have been performed on normalized, baseline subtracted spectra within the spectral region 1900–400 cm${}^{-1}$.

### 3.1. The Lollipop Chart

To summarize a large amount of data (746 spectra) in a visually interpretable form, the lollipop chart [35] has been employed. The lollipop chart functions identically to a normal bar plot, visually consisting of a line anchored at the x- or y-axis and a dot at the end to indicate the value. It illustrates the relationship between a numeric and a categorical variable. Lollipop charts are preferred when there is a lot of data to present, which can cause clutter when displayed as bars. The length of the bar represents (in our case) the magnitude of the distances between FTIR spectra collected for reference materials (C-4, PETN, or TNT) and the sample of residue after a controlled explosion using selected HEs. Distance functions are fundamental in essential data analysis as they measure the difference between two observations, in our case FTIR spectra. In this work, the Euclidean distance of the cumulative spectrum (ECS), typically used in spectrum search procedures and hierarchical cluster analysis, was found to be the most suitable distance measure based on the criteria presented in [36].

Figure 1 depicts an example of a lollipop chart calculated for FTIR spectra collected for samples that underwent controlled explosions using PETN. For clarity of presentation, two charts are shown: one containing all spectra, color-coded by the type of explosion (low- and high-order), and another with spectra collected only for high-order post-blast residues (this time, color-coding was used to differentiate the type of sample from which the residues were collected). As seen, this presentation can be considered as a summary of a database search, where a single unknown spectrum is searched against a reference database containing thousands of reference spectra. In our case, the database contains only one spectrum—collected for PETN; spectra for C-4 and TNT were also included in this analysis as the experimental spectra and the distances between them and PETN spectrum were calculated. This summary presents the data trend of spectra in a simple way. As expected, the spectral distances calculated for the low-order explosion spectra are significantly smaller, indicating that the chemical signature of PETN was fully preserved after the explosion. However, in some cases, the length of the bars for the high-order explosion spectra is shorter than that for the low-order. This can serve as additional evidence, as noted in [37] that undetonated residues can still be found even after a high-order explosion. As seen in Figure 1b, various values for distances have been recorded for high-order blast residues. The fact that each material used in controlled explosions has its own unique spectral signature, corresponding to its composition, may explain these differences. This signature may overlap with the characteristic bands of HE materials. Given that the suitability of materials in retaining explosive residues has been poorly investigated [38], our work also focuses on this topic. As seen in Figure 1, metal, glass, and even wood appear to be suitable materials for preserving explosive residues, as their distances to PETN are shorter than those to C-4 and TNT.

### 3.2. LDA-PCA

Our ultimate objective was to develop a model for identifying HE traces in analyzed FTIR spectra. For the initial study, linear discriminant analysis (LDA) was chosen as LDA aims to maximize the between-class variance while minimizing the within-class variance through the application of linear discriminant functions [39]. As a first step, the spectra were divided into two subsets: a training set to build the model and a testing set to evaluate the predictive model using a cross-validation procedure. In our work, we decided to use spectra collected after high-order explosions as a training set (193 spectra), and the spectra collected from the remnants after low-order explosions as a test set (553 spectra). Overall, 100% of the variabilities in the system were explained by only two first LDs, but additionally, the procedure was ended with the warning that all variables are collinearly dependent. It is very typical for the spectroscopic experiment results that the number of input variables (features) greater than the number of training samples leads to over-fitting. Our data have this characteristic, with 1501 variables versus 193 spectra. In order to remove the over-fitting and collinearity of highly correlated spectral data, further reduction of a number of variables was required. The PCA algorithm was implemented for data dimensionality reduction. The PCA model was built by applying the normalized and scaled FTIR spectra within the region 1900–400 cm${}^{-1}$. PCA runs an orthogonal transformation to convert data (of possibly correlated variables) into a batch of unique variables called principal components (PCs) that successively augment variance. It is confirmed to be a basic and efficient dimensionality reduction method for spectroscopic data. New, uncorrelated variables PCs are, by design, orthogonal and are ordered in such a way that the first carries the most variance of the system, the second most of the remaining variance, and so on. Before implementing the PCA algorithm, the dataset features were centered by excluding the mean and scaled. The first principal component (PC1) explained 28.9% of the variance, the second 21.6%, and the third 18.9%. The cumulative variance explained by PC4 to PC7 was relatively high, around 18.1%. The total variance explained by the first three principal components was 69.4% for our datasets. The validation of the minimum number of PCs that explain the majority of the system variability is one of the major challenges when applying PCA to a dataset and it depends on the particular problem [40]. There is no standard approach for selecting the appropriate number of PCs. One approach is to study a scree plot and search for “elbow” to establish the correct amount of PCs. Unfortunately, very often the plot has no clear “elbow” feature. Kaiser’s rule [41] suggests leaving the PCs with eigenvalues above 1, but in this case the number of factors extracted is usually about one third the number of variables in the original dataset regardless of whether many of the additional factors are noise. Cangelosi and Goriely [42] summarized the standard rules of thumb to detect the number of components in a study of component retention in PCA with application to cDNA microarray data. In this study, we decided to retain those PCs that cumulatively explain at least the 99% of the overall variance. Figure 2 shows that this condition is achieved when we choose to keep the first 21 PCs. Then, our new projected data consist of 746 spectra (observations), each with 21 features (variables).

#### 3.2.1. $COS^{2}$

To emphasize the highly correlated nature of the system analyzed in this work, the plot of the quality of representation for the original variables in the PC1-PC2 space is shown in Figure 3.

This plot, also known as the variable correlation plot, presents the relationship between all original variables (wavenumbers in this case). Positively correlated variables are gathered closely, while negatively correlated variables are located on opposite sectors of the plot’s origin. The parameter $cos^{2}$ (squared cosine or squared coordinates) was used in Figure 3 for color-coding the arrows representing the variables. A high value of $cos^{2}$ suggests a satisfactory representation of the variable in the principal component space. If the variable is completely expressed by solely the first two principal components (PC1 and PC2), the sum of the $cos^{2}$ for these two PCs is equal to 1 and the variables are positioned on the circle of correlations. For some variables, more than two components are needed to accurately describe the data, and in these cases the variables are positioned inside the circle of correlations. In Figure 3, variables with low $cos^{2}$ values are colored white, variables with medium values of $cos^{2}$ are colored blue, and eventually, variables with high $cos^{2}$ values are colored red. This type of plot is not usually very useful for spectroscopic data, since many of the original variables (wavenumbers) are highly correlated (being part of the same absorption band). However, it shows that the analyzed system is very complex and more than two PCs are required to fully explain its variability.

If, instead of the variables, the individual observations are plotted in the first two PCs’ space, the presentation becomes more intuitive. In this case, observations (spectra) similar to each other are grouped in the plot. Additionally, the individuals are color-coded by their contribution, calculated by dividing the respective principal component value of the first observation by the sum of the square roots of all observations. This shows how much each observation contributes to each principal component. As seen in Figure 4, three dominant groups can be identified. To check if these groups are connected to the explosive materials used during the blasts, another color-coding was applied. Each point was labeled according to the type of HE material used (C-4, PETN, or TNT) and the order of the explosion (low- or high-). For spectra collected from low-order explosions, a distinct separation is observed for all HEs. For high-order explosions with C-4, some of the spectra are mixed with spectra collected after high-order explosions with TNT, while TNT spectra are mixed with PETN spectra (Figure 4).

#### 3.2.2. Partition Plot LDA

The discussed example illustrates that the first two PCs do not fully explain the variability of the system. To determine if other combinations of two PCs might provide better discrimination, partition plots of 2D principal component space were calculated using LDA algorithms. LDA manipulates data to break down prognosticator variables into categorical regions with linear boundaries. LDA orders dimensions based on how much segregation each group gains, enlarging the difference and diminishing the overlap of clusters. LDA can be regarded as the supervised method of figuring out the directions (linear discriminants) that describe the axes in order to maximize separation between different classes. Figure 5 presents the partition plots with pairs from PC1 to PC4. The colored areas represent the limits of each classification region, with yellow for PETN, orange for TNT, and blue for C-4. The observations are marked with the corresponding letters (P for PETN, T for TNT, and C for C-4) and located in the appropriate colored region, which indicates their predicted membership by LDA. Black letters indicate that the sample was correctly allocated, while red letters show that the sample was misclassified by the model. The yellow dots on the plots represent the mean value for each group. As can be seen, most of the observations were correctly classified for the region corresponding to the PETN, TNT and C-4 category only in PC1 vs. PC2 space; indeed, the apparent error rate calculated for these dimensions has the lowest value in the comparison to errors obtained for the other combinations of dimensions PC1 to PC4.

#### 3.2.3. Final LDA-PCA Model

As previously demonstrated, the first 21 PCs can explain at least 99% of the overall variance. To improve the discrimination rate, linear discriminant analysis (LDA) was performed on these 21 PCs. The model was built using the training set of 193 spectra collected after high-order explosions and was evaluated using leave-one-out cross-validation. This validation process works by leaving out one observation and verifying the classification function using the remaining observations. This process is repeated for each observation, allowing the function of the other observations to allocate each observation. In our case, this procedure resulted in a cross-validation accuracy of 100%. The LDA model, built on spectra from high-order explosions, correctly classified all 553 spectra from low-order explosions, resulting in 100% testing accuracy.

As seen in Figure 6, a scatter plot of all observations in the training dataset for the first two values of linear discriminant functions shows that the training set (built on high-order explosion spectra) nicely overlaps with the points belonging to the test set (low-order spectra). Both sets are clustered in the first two linear discriminant functions space according to the type of explosive used for low- or high-order blasts. The percentage of separation achieved by LD1 is 80.3%, and 19.7% in the case of LD2. Additionally, probability density functions (PDF) have been calculated to check their distributions over the range of values that a variable can take. PDF (presented in Figure 6 as the margin plots separately for each HE material) for LD1 shows a clear separation between TNT and C-4, and some overlapping between PETN and C-4. This overlapping is well-resolved when PDF for LD2 is considered.

## 4. Conclusions

Numerous studies have shown that FTIR spectroscopy is a powerful method for classifying various samples based on differences in their chemical content. It is a straightforward analysis in terms of pure chemicals, but challenging if the analyzed samples are mixtures. In the present study, different types of materials, such as aluminum, glass, fabric, wood, steel, cardboard, plastic, and cotton, that were subjected to containment explosions were analyzed using FTIR spectroscopy. The aim was to detect traces of residues on their surfaces after high- and low-order explosions with three common HE materials. It is evident that a high amount of non-reacted HE material could contribute to the direct analysis. FTIR spectra collected for residues after low-order blasts were almost identical to the reference spectra. In the case of low-order explosions, practically all types of materials acted as convincing objects for evidence recovery. For high-order explosions, metal, glass, and even wood were found to be suitable materials for preserving explosive residues. Studying post-blast residues is a challenging task, as the samples typically contain only a small amount of unreacted explosive material mixed with reaction products and some inert material. These small amounts of HE materials present in FTIR spectra manifest themselves in the form of tiny peaks, sitting on the humps of wide absorption bands belonging to the characteristic spectral signatures of samples used as catchers; very often, overlapping the characteristic absorbance bands for explosives makes their identification almost impossible. Therefore, special methods must be used to ensure the reliability of the conclusions drawn from the measurements. Careful pre-processing of spectral data (baseline correction and normalization) was required to maximize positive identification. During the analysis, all possible external influences were eliminated. Various multivariate dimension reduction and clustering techniques were tested on a dataset of FTIR spectra collected from the debris after low- and high-order blasts. The most successful approach for identifying the high-energy material used in the blast was a combination of PCA for dimension reduction and LDA for building the discrimination model. PCA reduces the dimensionality by focusing on features with the most variation, while LDA is focused on maximizing the separability among known categories, in this case, groups of spectra collected after blasts with three common HEs: PETN, C-4, and TNT. The spectra collected for residues after high-order blasts were more challenging to analyze, so we deliberately used them as the training set in LDA. The LDA-PCA model revealed a sensitivity and specificity of 100% for the three HE groups. The percentage of between-class variance described by the first and second linear discriminant functions was 80.3% and 19.7%, respectively. LDA was executed on the first 21 PCs, and the cross-validation accuracy for our model was 100%. We can conclude that LDA-PCA is an adequate statistical method for classifying FTIR spectra measured from post-blast residues. This technique can be used to construct discriminant functions that allow for classifying new observations. All data manipulation and visualization of results were performed in R, an open-source environment for statistical computing. Based on our experiments, we confirmed that explosive materials can be found even in post-blast residues following high-order detonations. According to the literature [4], this is not uncommon, but little is known about the mechanism of their survival. We believe that by using proper classification techniques, the analysis of FTIR spectra of post-blast residues becomes more effective and significantly improves the successful identification of explosive materials in analyzed samples. In summary, this study demonstrates the potential of FTIR spectroscopy in identifying explosive residues on various types of materials following high- and low-order explosions. The findings of this study highlight the challenges of post-blast residue analysis (complex spectra, sample preparation) and the need for careful data pre-processing and advanced statistical methods for the successful identification of explosive materials.

Future research work for the classification of residues after high and low-order explosions using machine learning techniques on Fourier Transform Infrared (FTIR) spectra could include:Integration of additional analytical techniques: combining FTIR spectroscopy with other analytical techniques such as Raman spectroscopy or mass spectrometry could provide a more comprehensive understanding of the chemical composition of the residues and lead to improved classification results.Data augmentation and expansion: increasing the size and diversity of the FTIR spectral data used for training machine learning algorithms could lead to improved classification results, especially in cases where the number of samples is limited.Residue characterization: further research into the chemical and physical properties of residues produced by different types of explosions could provide additional information to improve the accuracy of classification results.Real-world applications: research could focus on the practical applications of FTIR spectroscopy and machine learning techniques for residue classification in real-world scenarios. This could include evaluating the effectiveness of the methods in different environments and under different conditions.

Overall, there is great potential for future research to improve the classification of residues after high- and low-order explosions using machine learning techniques on Fourier Transform Infrared (FTIR) spectra.

## Figures and Tables

**Figure 1 molecules-28-02233-f001:**
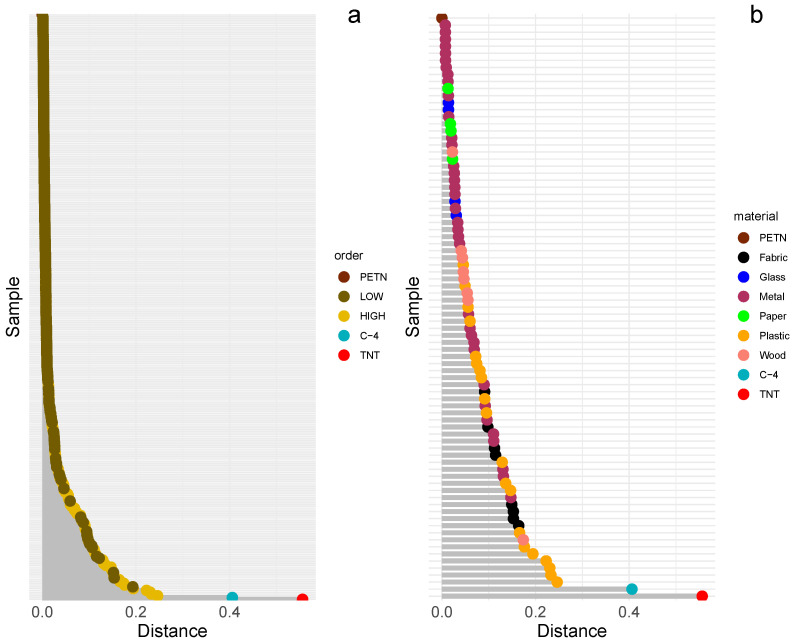
The lollipop charts for FTIR spectra taken for post-blast residue after the controlled explosion with the usage of PETN. (**a**) dots (spectra) are color-coded by the type of explosion (low- and high-order); (**b**) color denotes the type of material from which the samples were taken after the high-order explosion.

**Figure 2 molecules-28-02233-f002:**
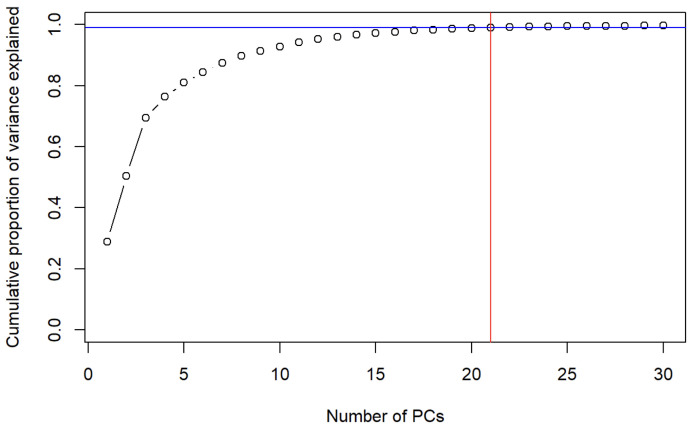
Cumulative proportion of variance explained.

**Figure 3 molecules-28-02233-f003:**
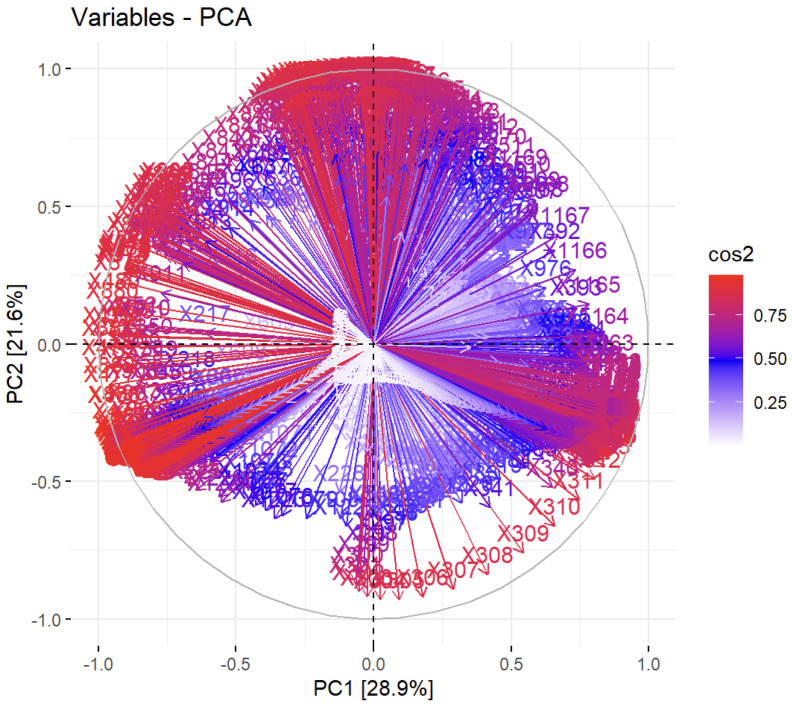
Plot of the quality of representation for variables on the factor map.

**Figure 4 molecules-28-02233-f004:**
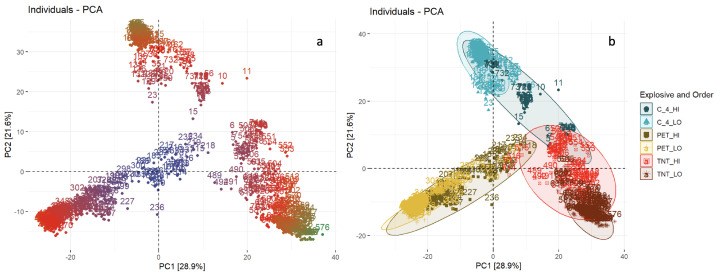
Principal component analysis (PCA). The scatter plot in PC1/PC2 space; each point corresponds to one spectrum and the points are colored by: the contribution of each observation to PC (**a**) and explosive material and order of explosions (**b**).

**Figure 5 molecules-28-02233-f005:**
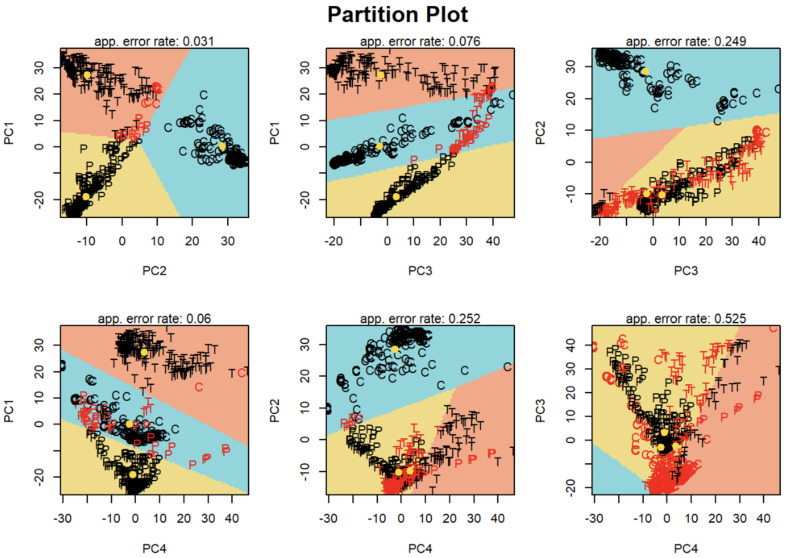
Partition plots resulting from the linear discriminant analysis in the PCs space (taking into linear discriminant analysis only pairs for PC1 to PC4). The color regions represent the limits of each classification area defined by the LDA model (blue = C-4, yellow = PETN, orange = TNT).

**Figure 6 molecules-28-02233-f006:**
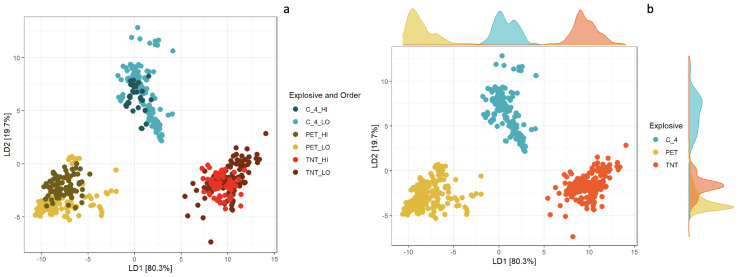
Discrimination plot of all observations obtained from the results of LDA model developed using training set (193 spectra (observations): C-4-, PETN-, TNT-high order) on first 21 PCs (**a**). Additionally, probability density functions (PDF) for each explosive are presented as the margin plots (**b**).

## Data Availability

The data presented in this study are available on request from the corresponding author.

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
