# Peer review of "Classification of the Residues after High and Low Order Explosions Using Machine Learning Techniques on Fourier Transform Infrared (FTIR) Spectra"

_molecules, 2023, doi:10.3390/molecules28052233_

Round 1

Reviewer 1 Report

Title: Classification Of The Residues After High And Low Order Explosions Using Machine Learning Techniques On Fourier Transform Infrared (FTIR) Spectra.

Journal: Molecules (ISSN 1420-3049)

Manuscript ID: molecules-2166521

After carefully evaluation. I am pleased to send you some comments. Please consider these suggestions as listed below to prepare the article again.  

  1. The title seems ok.
  2. The abstract seems very OK. Please add introductory lines in the beginning.
  3. Keywords are very weird. Please provide ONLY 5 keywords.
  4. Research gap should be delivered on more clear way with directed necessity for the future research work.
  5. Introduction section must be written on more quality way, i.e., more up-to-date references addressed.
  6. The novelty of the work must be clearly addressed and discussed, compare previous research with existing research findings and highlight novelty.
  7. What is the main challenge?
  8. What is problem statement? State clearly.
  9. Do not use lumpy references. Maximum should be 2 or 3. Please revise your paper accordingly since some issue occurs on several spots in the paper.
  10. Please check the abbreviations of words throughout the article. All should be consistent.
  11. The main objective of the work must be written on the clearer and more concise way at the end of introduction section.
  12. There are several irrelevant reference please take a strong revision in this section.
  13. Please provide space between number and units. Please revise your paper accordingly since some issue occurs on several spots in the paper.
  14. To meet the journal standard, author should add a comparative profile.
  15. Author should add a separate section with name challenges and future perfectives.
  16. Conclusion section is missing some perspective related to the future research work, quantify main research findings, and highlight relevance of the work with respect to the field aspect.
  17. To avoid grammar and linguistic mistakes, MAJOR level English language should be thoroughly checked. Please revise your paper accordingly since several language issue occurs on several spots in the paper.
  18. Reference formatting need carefully revision. All must be consistent in one format. Please follow the journal guidelines.  

 Decision = Major revision. It was tough for me to read and go through, but I was able to make a comments for improvement. Please put forth your best efforts and revised it. The idea is good so, I recommend a chance to revise it.

Reviewer 2 Report

The article presents the results of the analysis of post-blast residues by FTIR to detect the explosive type, which  is a quite challenging task. But the authors successfully solved this task.

Fig.1  needs revision. For the left chart the Y-axis is not readable. For X-axis the distance must be signed for example - meter.

I recommend to accept the article but for the readers of Molecules there are to much pecular mathematics but not chemistry or crystallography

Reviewer 3 Report

see the attachment

Round 2

Reviewer 1 Report

Accepted.